materials science/environmental chemistry

metal-organic framework, Fe catalyst, catalyst preparation, organic pollutant in water, advanced oxidation

**Authors for correspondence:**
Yanyan Xi
e-mail: xiyy@upc.edu.cn
Xufeng Lin
e-mail: hatrick2009@upc.edu.cn

This article has been edited by the Royal Society of Chemistry, including the commissioning, peer review process and editorial aspects up to the point of acceptance.

# UiO-66-supported Fe catalyst: a vapour deposition preparation method and its superior catalytic performance for removal of organic pollutants in water

Huimin Zhuang[1,2], Bili Chen[2], Wenjin Cai[2], Yanyan Xi[1,3], Tianxu Ye[1,2], Chuangye Wang[1,2] and Xufeng Lin[1,2]

[1]State Key Laboratory of Heavy Oil Processing, [2]College of Science, and [3]College of Chemical Engineering, China University of Petroleum (East China), Qingdao 266580, People's Republic of China

XL, 0000-0003-0256-9092

A vapour deposition (VD) method was established for preparation of the UiO-66-supported Fe (Fe/UiO-66) catalyst, which provided the first case of the metal-organic framework (MOF)-supported Fe catalyst prepared by using the vapour-based method. The Fe loading was around 7.0–8.5 wt% under the present preparation conditions. The crystal structure of UiO-66 was not obviously influenced by the Fe loading, while the surface area significantly decreased, implicating most of the Fe components resided in the pores on UiO-66. The results for the methyl orange (MO) removal tests showed that MO in aqueous solution can be removed by UiO-66 by adsorption, and in contrast, it can be oxidized by $H_2O_2$ with the catalysis of Fe/UiO-66. Further catalytic tests showed that Fe/UiO-66 was rather effective to catalyse the oxidation of benzene derivatives like aniline in water in terms of chemical oxygen demand (COD) removal efficiency. The catalytic test results for Fe/UiO-66 were compared to those of $Fe/Al_2O_3$ with the same Fe loading and to the catalysts reported in the literature. This paper provides a general strategy for VD preparation of MOF-supported Fe catalyst on the one hand, and new catalysts for removing organic pollutants from water, on the other hand.

## 1. Introduction

Water contamination by organic pollutants, especially toxic ones, is one of the most serious environmental problems today.

Developing new techniques for water treatment is an increasingly urgent issue for humankind due to the increasing demand for clean water [1,2]. The organic pollutants in water can be removed by various techniques such as physical, chemical and biological methods. Among these techniques, the Fenton reactions in particular and the advanced oxidation processes (AOP) in general are considered as promising and advantageous techniques owing to their strong oxidative capacity, wide applicability and mild reaction conditions [3,4]. They are often used when the disposed organic pollutants are poorly bio-degradable. In a classic Fenton reaction using a homogeneous $Fe^{3+}$ catalyst, hydroxyl radical ($\cdot$OH), as a reactive intermediate, can be generated; it can non-selectively attack almost all the organic pollutants and finally mineralize them into less harmful small molecules or even $CO_2$ and $H_2O$ [5–8]. However, the classic Fenton reaction suffers from several drawbacks including (i) rigorous operating pH range, (ii) introduction of a large amount of Fe salts and thus requirement of further treatment for Fe-containing liquid/sludge, and (iii) the inefficient production of $\cdot$OH leading to high operating costs especially in the case of a large scale application [6,9,10]. Therefore, great research efforts have been taken to develop heterogeneous solid catalysts to overcome the above-mentioned drawbacks of homogeneous $Fe^{3+}$ catalytic systems.

For the sake of catalytic removal of organic pollutants in aqueous solution, various kinds of Fe-based heterogeneous catalysts have been developed for Fenton-like reactions so far, such as $Fe_3O_4$ [11], $Fe_2O_3$/$SiO_2$ [12], $\beta$-FeOOH/rGO [13], $Fe_2O_3$/MWCNT [14], Fe/ZSM-5 [15] and ascorbic acid/Fe@$Fe_2O_3$ [16], etc. For example, approximately 80% phenol can be removed at 80°C after reaction with $H_2O_2$ in water for 200 min by using the $Fe_2O_3$/MWCNTs catalyst. However, as a whole, the catalytic performance of the presently reported heterogeneous Fe catalysts in the literature still needs further improvement in order to meet the requirements of commercial application. This calls for the development of new preparation methods for heterogeneous metal catalysts in general and supported Fe catalysts in particular.

Metal-organic frameworks (MOFs) belong to a type of inorganic–organic hybrid porous crystalline materials possessing strictly three-dimensional structures and constructed by metal-containing nodes connected by various organic bridges [17–21]. The MOF-based materials have been widely applied in various areas such as gas storage [22] and heterogeneous catalysis [1,22–27]. In particular, in recent years, some Fe-based MOFs have been studied in heterogeneous Fenton degradation of organic pollutants. For instance, Jiang *et al*. [1] employed MIL-53(Fe) to catalyse the decomposition of Rhodamine B in the presence of $H_2O_2$. This catalyst could completely decompose the $10\,\text{mg}\,l^{-1}$ Rhodamine B with a certain amount of $H_2O_2$ under a visible light irradiation for less than 50 min. Zhao *et al*. [26] proposed that Fe(II)@MIL-100(Fe) can be used for heterogeneous Fenton degradation of high concentration methylene blue. Cai *et al*. [27] reported that the yolk/shell Pd@$Fe_3O_4$@MOF catalysts were highly efficient to eliminate chlorophenols and phenols due to a continuous generation of $\cdot$OH. It is generally believed that the Fe-based MOF materials possess not only abundant exposed active sites [Fe(III)] for heterogeneous Fenton reaction but also have a favourable accessibility for reactants to active sites, making them promising Fenton-like catalysts for wastewater treatment. However, most MOF materials suffer from low thermal and/or hydrolytic stability [28], which may lead to metal active sites being blocked by the organic linker or solvent. In addition, most MOF materials are quite expensive. Although thousands of types of MOF have been reported, only very few of them are commercially available at present.

There are two strategies when using MOFs as catalysts in terms of catalytically active components [22–27]. The first strategy is that the MOF ingredients, especially their metal sites, are catalytically active. The second one is that MOF ingredients are not necessarily active for catalysis, and instead, the active components are introduced and resided on the surface of MOFs. Obviously, the second strategy allows people to have wider options in the selection of MOF type than the first one. This work aims at developing a preparation method for MOF-supported Fe catalysts with the following two advantages. First, the method can be rather easy to extend to different types of MOF as catalytic support. Second, the method is low-cost and environmentally-friendly. For this goal, the vapour deposition (VD) method can be a good candidate, based on our previous work [29] on Fe/$Al_2O_3$ preparation by a VD method. In ref. [29], we have reported a VD method for preparing alumina-supported Fe (Fe/$Al_2O_3$) catalyst using ferrocene as the Fe precursor. The Fe loading on the VD-Fe/$Al_2O_3$ catalyst was controllable within the range of 0–7 wt% by varying the deposition temperature and the flow rate of carrier gas. In this context, an interesting question arises as to whether or not the same preparation method can be used for preparing MOF-supported Fe catalyst.

This paper reports a simple, carrier-gas free VD method for preparation of MOF-supported Fe catalyst, and the application of the prepared Fe catalyst in catalytic oxidation of various organic compounds in water. In this work, UiO-66 was selected as an MOF candidate, due to high surface

area and unprecedented stability [30]. To the best of our knowledge, this paper provides the first case of MOF-supported Fe catalyst prepared by the VD method. Interestingly moreover, the obtained Fe catalyst presented superior performance in catalytic oxidation of aqueous organic compound compared to Fe-containing MOFs reported in the literature [1,26,27,31] and compared to a traditional Fe/Al$_2$O$_3$ catalyst.

# 2. Experimental section

## 2.1. Materials

4-carboxybenzene MOF (UiO-66) and 1,3,5-benzenetricarboxylic acid (H$_3$BTC, ≥99%) were provided by Beijing J&K Technology Co., Ltd. Ferrocene (C$_{10}$H$_{10}$Fe), Fe(NO$_3$)$_3$·9H$_2$O, methyl orange (MO, C$_{14}$H$_{14}$N$_3$NaO$_3$S), aqueous hydrogen peroxide solution (H$_2$O$_2$, 30 wt%), acetic acid (HAc), aniline, phenol, benzoic acid (BA), acetone and anhydrous sodium acetate (NaAc) were provided by Sinopharm Chemical Reagent Co, Ltd. All chemicals were of analytical grade and used without further treatment. The deionized (DI) water was used throughout the experiments.

## 2.2. Catalyst preparation

Fe/UiO-66 was synthesized with a VD method using a process revised from a previously reported VD method in our group [29]. Ferrocene was used as the Fe precursor and UiO-66 was used as the catalyst support. The whole preparation process contained the following steps. First, a watch glass containing 1.0 g UiO-66 powder was heated at 300°C in air for 2 h to remove the moisture and solvent by using a muffle furnace. This step is hereafter known as pre-calcination since it was performed before the VD step described later. Second, after cooling down, the watch glass containing precalcined UiO-66 was quickly placed in the centre of a Petri dish of an appropriate size, and then 1.0 g of ferrocene powder was spread evenly around the periphery of the watch glass and within the Petri dish. Then the Petri dish was covered firmly. The whole Petri dish set-up before the VD step can be depicted in figure 1*a*. The third step is the VD step. In this step, the whole Petri dish set-up was heated at a temperature (denoted as $T_d$) of 130–150°C for 2 h. Fourth, after cooling down, the Petri dish was uncovered, and the remaining ferrocene on the wall of glass vessels was removed. Then the Petri dish was covered again, as shown in figure 1*b*, and the whole Petri dish was calcined at 200°C for 1 h. Fifth, after cooling down, the Petri dish was uncovered as shown in figure 1*c*, and the whole Petri dish was calcined in air at 200°C for 4 h. Then the target catalyst, appeared in uniform yellow, was obtained, and is denoted as UiO-66 supported Fe catalyst, or simply Fe/UiO-66 in this paper. Since the above-described preparation method did not include the use of carrier gas, it was referred to as a carrier-gas free VD method.

Fe catalyst supported on γ-alumina (denoted as Fe/Al$_2$O$_3$) was prepared using a conventional equal-volume wetness impregnation method in general [32]. The Fe loading on Fe/Al$_2$O$_3$ was controlled to be the same as with Fe/UiO-66. The γ-alumina particles were prepared by a stripping method reported elsewhere [33].

## 2.3. Catalyst characterization

The crystal structure of a certain sample was evaluated by powder X-ray diffraction (XRD, D8 Advance, Germany) using Cu K$_\alpha$ radiation (λ = 0.15418 nm) at 40 kV, 30 mA, with a scan step of 0.02° and a 2θ ranging from 5 to 75°. The morphology and structure of the prepared catalysts were measured by a field emission scanning electron microscope (SEM, Hitachi SU3500). The energy-dispersive X-ray (EDX) mapping was performed on another SEM (Octane plus, Ametek) for the element distributions of Fe and Zr. The 77 K-N$_2$ adsorption–desorption isotherm of a certain sample was measured by using a Micromeritics ASAP2010 instrument. Then the specific surface area and pore-size distribution of this catalyst were calculated using the Brunauer–Emmett–Teller (BET) method for the adsorption branch and the Barrett–Joyner–Halenda (BJH) method for a desorption branch. The thermogravimetric (TG) analysis of a certain catalyst was carried out in an air flow of 50 ml min$^{-1}$ by using a HCT-1 thermal analytical instrument. The sample weight was approximately 10 mg and the typical heating rate was 10 K min$^{-1}$.

A certain catalyst was treated to form an aqueous solution before its Fe content was measured. In a typical treatment process, the catalyst (approx. 2 mg) was placed in a polytetrafluoro (PFT) beaker

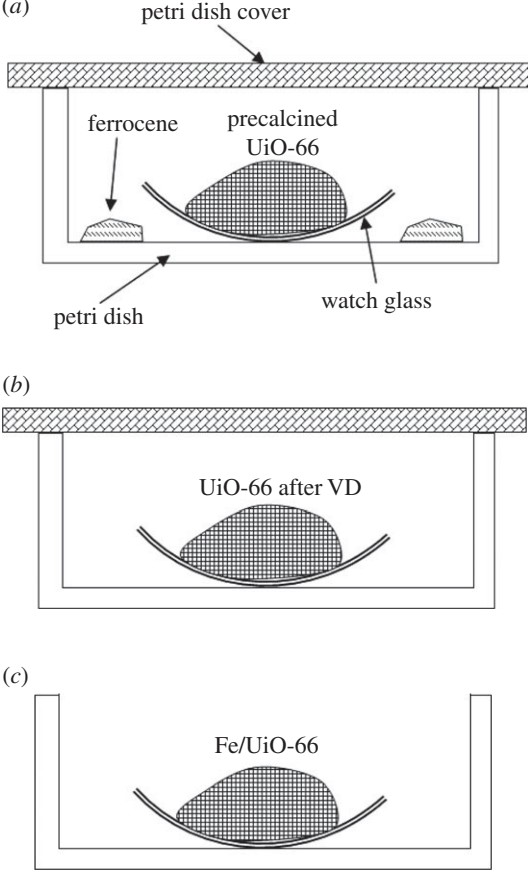

**Figure 1.** Simple schematic diagrams of the key steps for VD preparation of the Fe/UiO-66 catalyst. (*a*) The set-up immediately before and during the VD step, (*b*) the set-up for the first calcination after the VD step and (*c*) the set-up for the second calcination after the VD step. See the whole preparation process in detail in the text.

containing a small amount of DI water. Then concentrated nitric acid (approx. 67%, 4 ml), concentrated hydrofluoric acid (approx. 40%, 4 ml) and concentrated perchloric acid (approx. 71%, 0.5 ml) were added to the PFT beaker. This beaker was then heated, and during this heating process the solid sample was dissolved, accompanied by white smoke. After the white smoke was exhausted, 2 drops of hydrochloric acid (approx. 18 wt%) was added to dissolve the solid residue. The beaker was kept heated until a clear liquid solution was obtained and no solid could be observed. This liquid solution was transferred to a volumetric flask and diluted to 100.0 ml with DI water. The Fe concentration of the diluted solution was measured by atomic absorption spectroscopy by using a Contr AA700 spectrograph. The Fe content of the initial catalyst can be calculated from the Fe concentration value.

## 2.4. Catalytic oxidation of organic pollutants

The catalytic activity of Fe/UiO-66 for the advanced oxidation of aqueous organic pollutants was evaluated by using a batch reactor of approximately 250 ml flask. In a typical catalytic experiment, a 200 ml aqueous solution of a selected organic compound was added to the reactor. The option of organic compounds as the model organic pollutants was one of the following compounds, namely, MO ($40.0 \, \text{mg} \, \text{l}^{-1}$), aniline ($500 \, \text{mg} \, \text{l}^{-1}$), phenol ($500 \, \text{mg} \, \text{l}^{-1}$), BA ($600 \, \text{mg} \, \text{l}^{-1}$), acetone ($500 \, \text{mg} \, \text{l}^{-1}$), $H_3BTC$ ($1000 \, \text{mg} \, \text{l}^{-1}$). The temperature (typically being $45°C$ or $50°C$ as indicated in the following sections) of the reactor was controlled by water bath. When the reaction system maintained the temperature of interest, 2.0 ml $H_2O_2$ solution (30.0 wt%) was added to the above organic solution (the mixed solution had a $H_2O_2$ concentration of $0.087 \, \text{mol} \, \text{l}^{-1}$). Upon examination, no reaction between $H_2O_2$ with any of the above organic compounds can be observed without catalyst at the reaction temperature of interest. Fe/UiO-66 of 200 mg was added to the reactor to initiate the catalytic oxidation reaction, making the dosage of catalyst $1000 \, \text{mg} \, \text{l}^{-1}$.

**Table 1.** The dependence of Fe loading of the Fe/UiO-66 catalyst on the deposition temperature ($T_d$).

| $T_d$ (°C) | 130 | 140 | 150 |
|---|---|---|---|
| Fe loading (wt%) | 7.2 | 7.8 | 8.5 |

After a certain reaction time, 2.0 ml liquid sample was withdrawn from the reaction mixture, and was centrifuged to remove the trace amount of solid residue. Then this liquid sample was analysed by either a UV-2450 UV–vis spectrometer or a chemical oxygen demand (COD) analyzer (HACH DR1010). The UV–vis spectrometer was used only when the organic compound was MO, and the absorption of the liquid sample (after being diluted with the HAc-NaAc butter solution to a fixed volume) at the wavelength of 467 nm was recorded to obtain the concentration of MO. The removal rate of MO can be calculated by the decrease of MO adsorption of the sampled liquid over that of the unreacted MO solution. For the organic compound other than MO, the COD value of this liquid sample was analysed with the COD analyzer.

# 3. Results and discussion

## 3.1. About the vapour deposition preparation method for Fe/UiO-66

It is notable that the carrier-gas free VD method for preparing supported Fe catalyst described in §2.2 is reported for the first time in this paper. This VD method was rather different from the one reported previously for VD-Fe/Al$_2$O$_3$ [29]. The main reason to develop a new VD method instead of using the previous method was that fine powder (as UiO-66 or some zeolites appears) was difficult to pack in a way similar to the previous method. Except for the saving of carrier gas, the carrier-gas free VD method had advantages compared to the previous one in that, (i) several glass vessels took the place of a tube furnace, resulting in further cost-saving and easier operation, (ii) no treatment of exhaust gas was needed as a consequence of no carrier gas being used. After the deposition step, the combined two-step calcination (fourth and fifth steps described in §2.2) was similar for both VD methods. It should be noted that the combined two-step calcination instead of a single calcination step played a key role to convert physically adsorbed ferrocene into the final Fe components on the catalyst [29].

Table 1 shows that a Fe loading of 7.0–8.5 wt% can be achieved for Fe/UiO-66 at the $T_d$ of 130–150°C. Interestingly, as a comparison, on VD-Fe/Al$_2$O$_3$ prepared by the previous VD method, the Fe loading reached a maximum of around 5.5% at 130°C when the carrier gas flow rate was in the range of 20–40 sccm. The Fe loading decreased rapidly with the decrease of flow rate for VD-Fe/Al$_2$O$_3$. These trends reflected that the catalytic support in these two catalysts may play an important role in influencing the Fe loading. Given that the Fe loading was mainly determined by the amount of ferrocene physically adsorbed on the surface [29], the differences in the preparation parameters and in the Fe loadings for these two catalysts implicated that large surface area may be a key factor in the successful preparation of Fe/UiO-66 (*vide infra*). As it is well known that most MOF materials have a very large surface area [17–21], in principle, the carrier-gas free VD preparation method can be easily extended to other types of MOF as catalytic support.

## 3.2. Physiochemical properties of UiO-66 and Fe/UiO-66

During the course of establishing the catalyst preparation method, it was found that the deposition efficiency was rather poor if UiO-66 was not pre-calcined before VD. This can be accounted for the moisture contained in the pore of UiO-66. Considering that a pre-calcination process was necessary to remove moisture on the one hand and UiO-66 may be unstable at a too high temperature on the other hand, an explicit selection of the pre-calcination condition became a key issue for the successful catalyst preparation (see the first step in §2.2). To obtain an appropriate pre-calcination condition, the XRD patterns of pure UiO-66 under different calcination conditions were examined. At the calcination temperature of over 300°C, the XRD peaks decreased dramatically compared to the uncalcined UiO-66 (see in figure 2a). A calcination time of over 4 h can also lead to the decrease of the XRD peak of UiO-66. Therefore, the optimal pre-calcination condition of UiO-66 (used as carrier) before the VD step was 300°C for 2 h in the present stage. The BET surface area and pore size of the pre-calcined

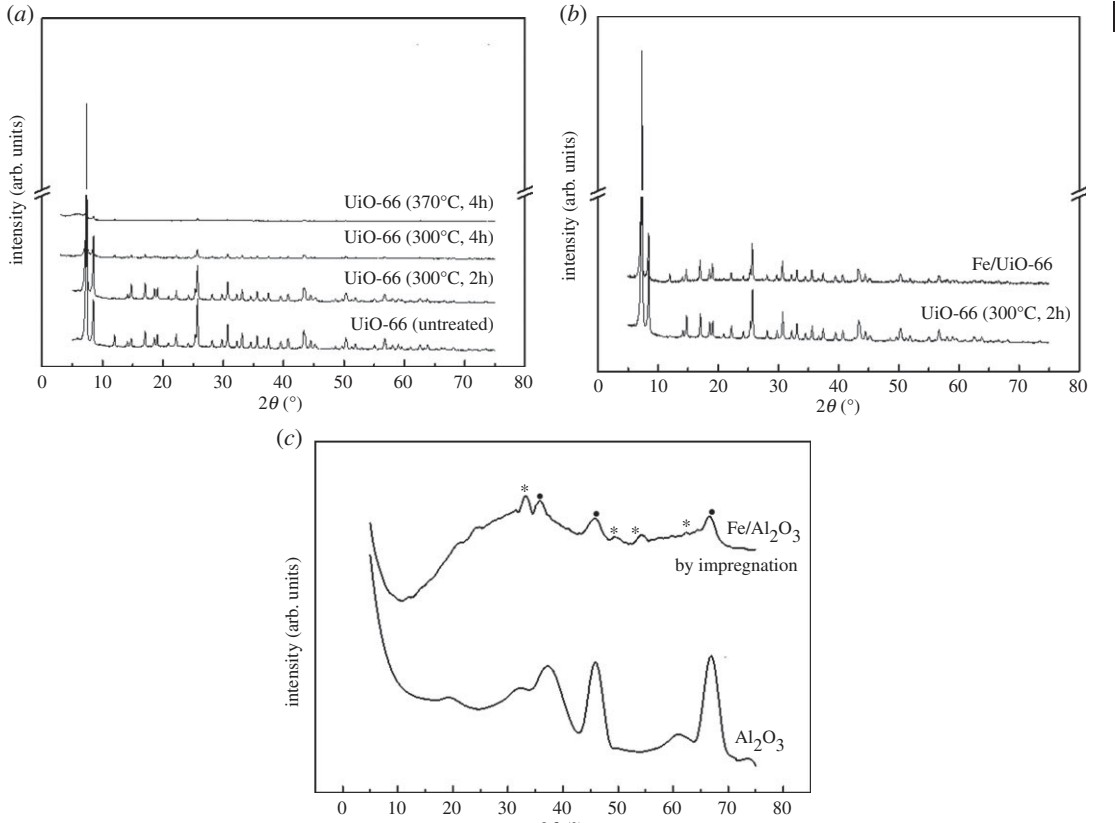

**Figure 2.** XRD patterns of (*a*) UiO-66 samples under different treatment conditions, (*b*) the Fe/UiO-66 ($T_d = 50°C$) catalyst compared to undeposited UiO-66 and (*c*) the $Al_2O_3$ support and the Fe/$Al_2O_3$ catalyst prepared by impregnation. Here Fe/$Al_2O_3$ had the same Fe loading (8.5 wt%) as Fe/UiO-66 in this case. In (*c*), the peaks marked with dots corresponded to the $Al_2O_3$ support, and the ones with asterisks (*) corresponded to $\alpha$-$Fe_2O_3$ according to the Joint Committee on Powder Diffraction Standards number of 00−033-0664.

**Table 2.** BET results of the untreated UiO-66, pre-calcined UiO-66 (300°C, 2 h), the Fe/UiO-66 catalyst, the $Al_2O_3$ support and the Fe/$Al_2O_3$ catalyst.

| samples | BET surface area $S_{BET}$ (m$^2$ g$^{-1}$) | total pore volume (cm$^3$ g$^{-1}$) | average pore diameter (nm) |
|---|---|---|---|
| UiO-66 (original) | 896 | 0.56 | 1.89 |
| UiO-66 (300°C, 2 h) | 895 | 0.51 | 1.99 |
| Fe/UiO-66 | 136 | 0.088 | 2.00 |
| $Al_2O_3$ | 331 | 1.10 | 6.70 |
| Fe/$Al_2O_3$ | 204 | 0.49 | 7.99 |

UiO-66 were almost the same as those of untreated UiO-66 (table 2), which further supported the rationale of the pre-calcination condition.

As shown in figure 2*b*, all peaks of Fe/UiO-66 ($T_d = 150°C$) matched well with those of the pre-calcinated UiO-66 material, indicating that the whole VD process did not lead to a large change in the crystal structure of UiO-66. Hereafter in this paper, only the results of Fe/UiO-66 with the $T_d$ of 150°C will be shown. The adjacent peaks were clearly divided and no impurity peaks were detected, indicating a good crystallization of the samples. A new weak peak was observed at around 12° which may be due to the formation of a small amount of Fe–C-containing compounds [26]. Given that the Fe loading was rather high (8.5 wt% at $T_d = 150°C$, table 1), the absence of the Fe oxide features in the XRD pattern of Fe/UiO-66 suggests that the Fe components could be rather uniformly dispersed

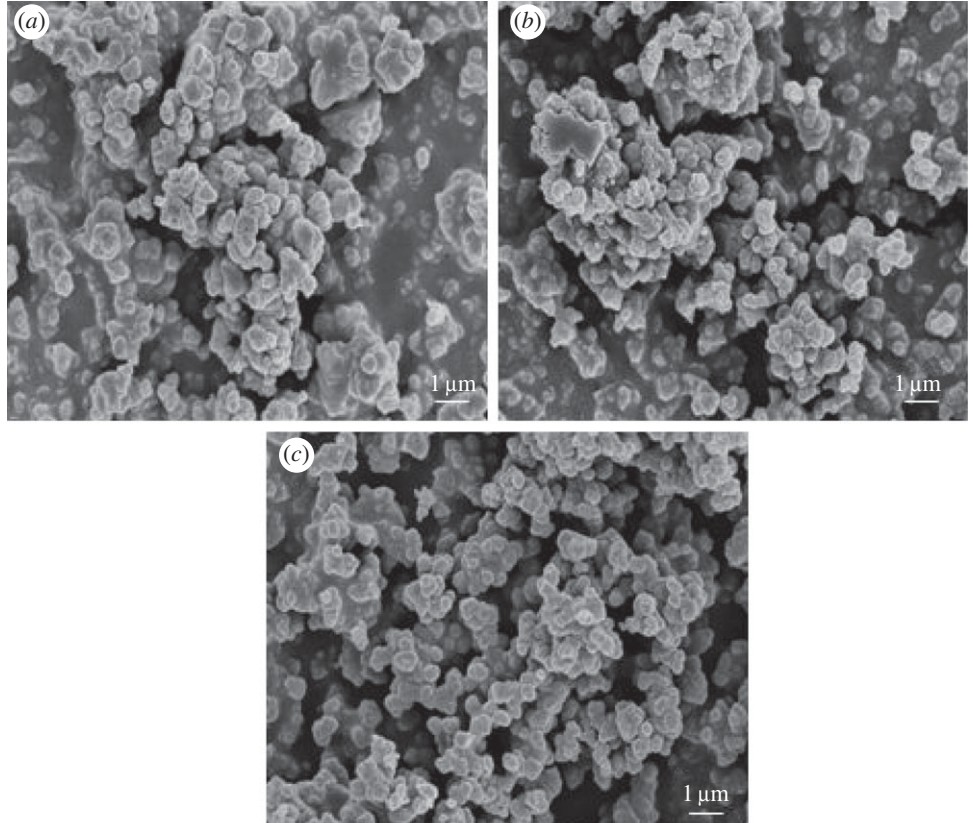

**Figure 3.** SEM images of (*a*) the original UiO-66 before pre-calcination, (*b*) UiO-66 after pre-calcination in air at 300°C for 2 h and (*c*) the Fe/UiO-66 catalyst.

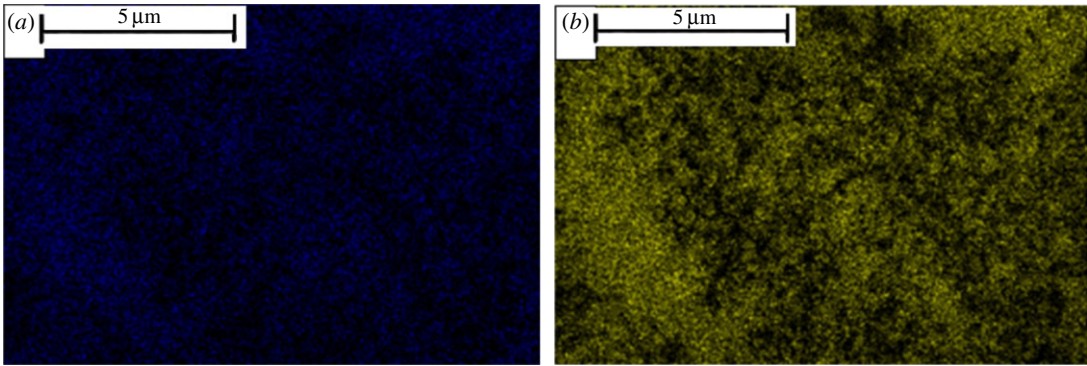

**Figure 4.** The EDX mapping of Fe/UiO-66 for (*a*) Fe element and (*b*) Zr element.

on the UiO-66 surface (mainly inner surface). As a comparison, an obvious $\alpha$-$Fe_2O_3$ feature can be identified for the Fe/$Al_2O_3$ catalyst prepared by impregnation with the same Fe loading as Fe/UiO-66 (figure 2*c* for the XRD patterns of $Al_2O_3$ and Fe/$Al_2O_3$ and the peak assignment).

Figure 3 shows the SEM images of UiO-66 and Fe/UiO-66. Inspection of figure 3*a*,*b* shows no obvious difference between the original UiO-66 sample and pre-calcined UiO-66, further supporting the rationality of the selection of pre-calcination condition (300°C for 2 h). Meanwhile, the images of the pre-calcined UiO-66 and the Fe/UiO-66 catalyst (figure 3*b* and *c*) also did not exhibit a significant change in the morphology, indicating that the Fe-component introduction process in the whole VD process did not notably alter the structure and morphology of UiO-66, partly owing to the stability of UiO-66 at a high temperature. The EDX mapping (figure 4) results showed that the Fe distribution of Fe/UiO-66 was highly consistent with the Zr distribution at the same region. Since the Zr distribution was determined by the UiO-66 framework, the EDX mapping results further support the uniform dispersion of the Fe element on the UiO-66 surface.

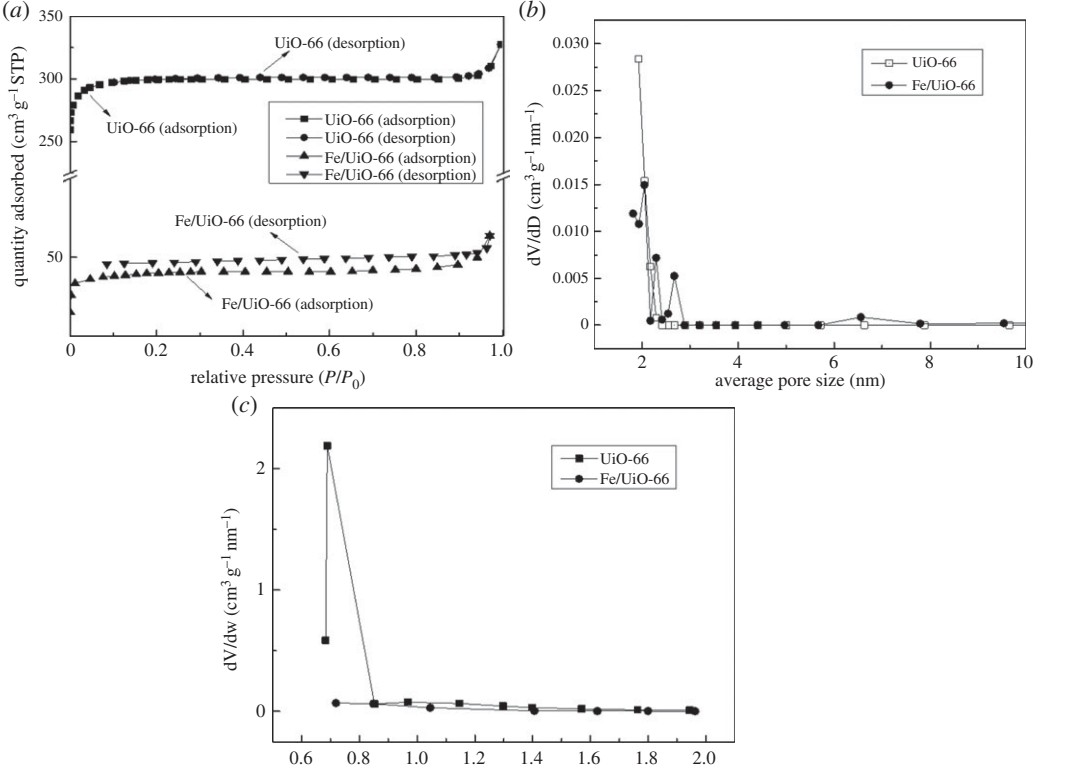

**Figure 5.** (a) 77 K-N$_2$ adsorption–desorption isotherms of UiO-66 and Fe/UiO-66, and (b) the BJH mesoporous size distribution and (c) the HK micropore size distribution of UiO-66 and Fe/UiO-66 derived from (a).

Figure 5 shows the N$_2$ adsorption–desorption isotherms and pore size distributions of UiO-66 and Fe/UiO-66. As shown in figure 5a, the isotherms of UiO-66 and Fe/UiO-66 displayed a mode of type I, which indicated that the materials mainly had microporous windows and possibly a small portion of mesoporous cages [34,35]. As derived from the N$_2$ adsorption data (table 2), the UiO-66 material possessed a high BET surface area of 896 m$^2$/g with a pore volume of 0.51 cm$^3$ g$^{-1}$, while the BET surface area and the pore volume of Fe/UiO-66 catalyst decreased dramatically compared to the UiO-66 case. This dramatic decrease reflected that the Fe components introduced by VD mainly resided within the pore of UiO-66 instead of sticking to the outer surface, since the Fe loading was moderately high (8.5 wt%). This hypothesis was supported by the similarity of XRD patterns between UiO-66 and Fe/UiO-66 (figure 2b) and also by the SEM images of these (figure 3). The micropore of UiO-66 was approximately 0.68 nm from the Horvath–Kawazoe analysis (figure 5c), which was absent for Fe/UiO-66, and further supported the above hypothesis. The pore size distributions of both UiO-66 and Fe/UiO-66 (figure 5b) had a maximum pore diameter of 2.0 nm, which was slightly larger than the kinetic diameter of the MO molecule (approx. 1.4 nm) [36]. The favourable structure characteristics could facilitate the contact with reactants in catalysis.

Figure 6 shows the TG curves of UiO-66 and Fe/UiO-66. The TG curves of the UiO-66 sample exhibited two weight loss processes between 50 and 580°C, which agreed with the results reported by Yang et al. [37]. The first weight loss of approximately 10% between 50 and 350°C corresponded to the release of guest H$_2$O and DMF molecules. The second weight loss of approximately 50% between 400 and 580°C may be ascribed to the release of the organic ligand, which led to the framework structure decomposition. After the decomposition, about 40% of the starting weight remained, corresponding to the formation of 6ZrO$_2$ obtained from the formula Zr$_6$O$_4$(OH)$_4$(CO$_2$)$_{12}$ as reported by Lillerud et al. [30]. The TG curve of the Fe/UiO-66 sample showed a similar shape to that of UiO-66. There were also two weight loss steps in the TG curve of Fe/UiO-66. The first weight loss of approximately 5% between 50 and 300°C can be ascribed to the loss of guest H$_2$O molecules. The second weight loss of 50% between 300 and 450°C can be also accounted for by the release of organic ligand and framework structure decomposition. In short, although the residing of the Fe components in the UiO-66 support decreased its thermal stability in a minor way, the Fe/UiO-66 still had a rather

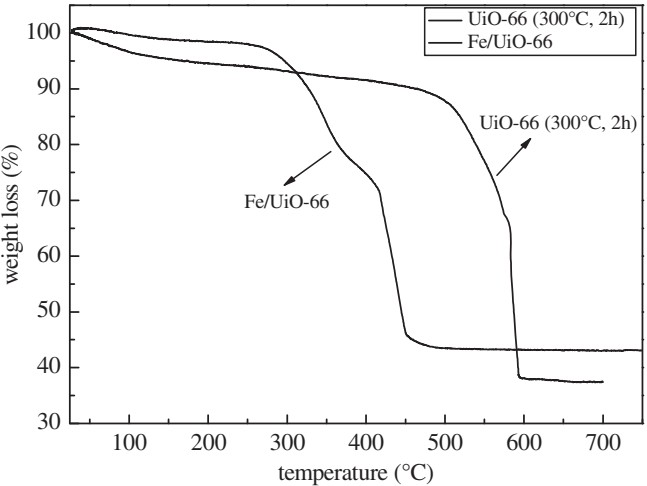

**Figure 6.** TGA curves of UiO-66 and Fe/UiO-66 in air atmosphere.

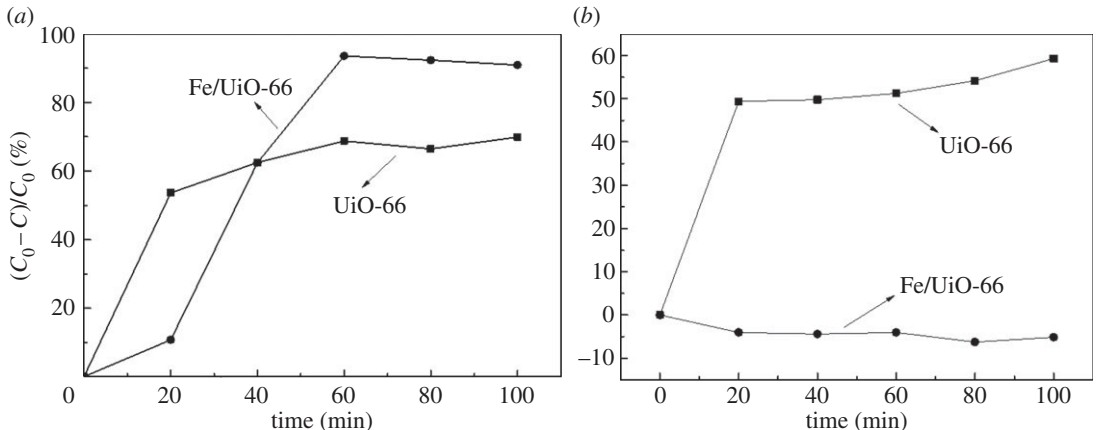

**Figure 7.** (a) The profile of the MO concentration (C) over its initial concentration ($C_0$) of the reaction mixture containing 40 mg $I^{-1}$ MO and 0.0087 mol $I^{-1}$ $H_2O_2$ in water. 1000 mg $I^{-1}$ UiO-66 or Fe/UiO-66 was used as the catalyst, and the reaction temperature was 45°C. (b) The same experiment as (a) with the only difference: $H_2O$ took the place of $H_2O_2$.

good thermal stability, which could make it hopeful for further catalytic application at high temperatures like 200–300°C.

## 3.3. Catalytic removal of methyl orange from water: catalytic oxidation versus adsorption

The success of the preparation of Fe/UiO-66 encouraged us to further explore its performance as a catalyst in general and for catalytic AOP processes in particular. Figure 7a shows the MO removal profile in the presence of $H_2O_2$ and a certain material of either pure UiO-66 or Fe/UiO-66. Within the first 40 min of reaction time, the removal rate of MO with UiO-66 was higher than that of Fe/UiO-66. However, the removal rate was obviously higher for the Fe/UiO-66 case than that for the pure UiO-66 case after 40 min. Given that UiO-66 had a much larger surface area and pore volume compared to Fe/UiO-66 (table 2), and adsorptions in general are quick processes compared to catalytic reactions, it was reasonable to hypothesize that the MO removal by UiO-66 may be mainly due to adsorption instead of catalytic reaction.

In order to verify the above hypothesis, the 'reaction test' experiments were also performed without $H_2O_2$ being added to the reactor. The results of such reaction tests are depicted in figure 7b. It can be noted that figure 7b differs from figure 7a, that is, water took the place of $H_2O_2$. Figure 7b shows that Fe/UiO-66 did not present observable activity in MO removal, reflecting that the MO removal by Fe/UiO-66 shown in figure 7a was due to catalytic oxidation reactions. Therefore in contrast to that, when pure UiO-66 was used, the MO removal rates were quite similar for $H_2O_2$ being present and being

absent. In short, Fe/UiO-66 can efficiently catalyse the oxidation reaction of MO with $H_2O_2$, and UiO-66 simply adsorbed MO from the aqueous solution [38].

Cui et al. [39] have reported the results about the removal of MO from water ($10\ mg\ l^{-1}$) with $0.3\ g\ l^{-1}$ Cu-MOFs ($[Cu_2(btec)(btx)_{1.5}]_n$) as catalyst. The reaction temperature they used was the same as that used in this work; however, the solution pH was adjusted to a mild acidic value (approx. 3) during their reaction tests. The MO removal efficiency was approximately 83% within 60 min. Similar MO removal efficiency was achieved when the catalysts were Co(II)-MOFs [40,41]. As a comparison, the removal efficiency of approximately 93% was achieved with 60 min (figure 7a) in this work. Overall, the Fe/UiO-66 catalyst in this work showed a comparable or better performance for the MO removal compared to the representative results in the literature.

More interestingly, no pH adjustment was required in our catalytic system, further saving the operating cost. As is known [39–41], catalytic removal of dyes in general and of MO in particular from water are always performed in moderately acidic solutions (typical pH of 3–4) instead of nearly neutral solutions in order to increase the removal efficiency. Thus the pH adjustment of the MO containing solutions was often seen in the literature. However, the acidity has obvious disadvantages in terms of practical application, for example, increase of the operating cost, and possible corrosion of metal containers resulting in the need for further treatment. From this point of view, it can be of great significance for a catalytic system using nearly neutral solutions. This was achieved in this work by using the Fe/UiO-66 catalyst.

## 3.4. Catalytic removal of other organic compounds from water

The effectiveness of the catalytic removal of MO from water by Fe/UiO-66 encouraged us to further explore the applicability of this catalyst when applied to other organic pollutants. As is well known, COD is a key index which reflects the degree of water pollution caused by organic contaminants. Hereafter in this paper, the COD removal percentage is used to represent the catalytic removal efficiency of Fe/UiO-66 for organic compound oxidized by $H_2O_2$. The reaction temperature was 50°C in the catalytic test results described in this section.

### 3.4.1 Benzene derivatives

Benzene derivatives belong to one of the most frequently occurring water-pollution sources in industry. In particular, aniline, as a useful chemical in industry, is a highly toxic compound. The removal of aniline is an important topic in the field of environmental protection [42,43]. Figure 8a shows the COD removal percentage of the aniline and $H_2O_2$ aqueous solution mixture with the catalyst of Fe/UiO-66 and with the catalyst of $Fe/Al_2O_3$, respectively. It is notable that the concentration of aniline was rather high ($500\ mg\ l^{-1}$) in this work. As can be seen in figure 8a, approximately 80% COD of the solution was removed with the Fe/UiO-66 catalyst within 180 min, while the percentage of COD removal was quite poor (less than 5%) when $Fe/Al_2O_3$ (having the same Fe loading as Fe/UiO-66) was used as the catalyst. Such comparison clearly showed the superior catalytic performance of Fe/UiO-66 for oxidative removal of aniline. Interestingly however, the BET surface area of Fe/UiO-66 was even smaller than that of $Fe/Al_2O_3$ (table 2). Recall the XRD patterns of UiO-66, Fe/UiO-66 and $Fe/Al_2O_3$; this superior catalytic performance can be partly accounted for the high dispersion of the Fe components on the UiO-66 support.

The COD removal efficiency of aniline by Fe/UiO-66 was also comparable to the Fe/ZSM-5 catalyst reported by Zhu et al. [43]. Inspection of the catalytic test parameters in this work and those in [43] shows that the aniline concentration in the former case was 2.5 times larger than that in the latter case, and the dosage of catalyst in the former was only 1/6 of the latter. Although the COD removal percentage was larger (92.5%, 120 min) for Fe/ZSM-5 than for Fe/UiO-66 (72.0%, 120 min), the overall catalytic performances of these two catalysts were quite comparable. Moreover, as described in §3.3 for the MO removal results, the catalytic system for aniline removal in this work did not require pH adjustment, while the condition of pH = 4 can be seen for the Fe/ZSM-5 catalytic system [43].

Treatment of phenol-polluted water is also a hot topic in environmental science and technology [44]. $Fe/Al_2O_3$ was more effective in removing phenol (figure 8b) than removing aniline (figure 8a). However, Fe/UiO-66 still obviously outperformed $Fe/Al_2O_3$ in terms of phenol removal efficiency. For example, the percentage of COD removal was 78.7% by Fe/UiO-66 at the reaction time of 90 min, being also twice that given by $Fe/Al_2O_3$. Sohrabi et al. [44] have used $5\ g\ l^{-1}$ Fe/clinoptilolite (Fe loading of 10 wt% larger than 8.5 wt% in this work) to catalyse the oxidative removal of phenol ($100\ mg\ l^{-1}$) by

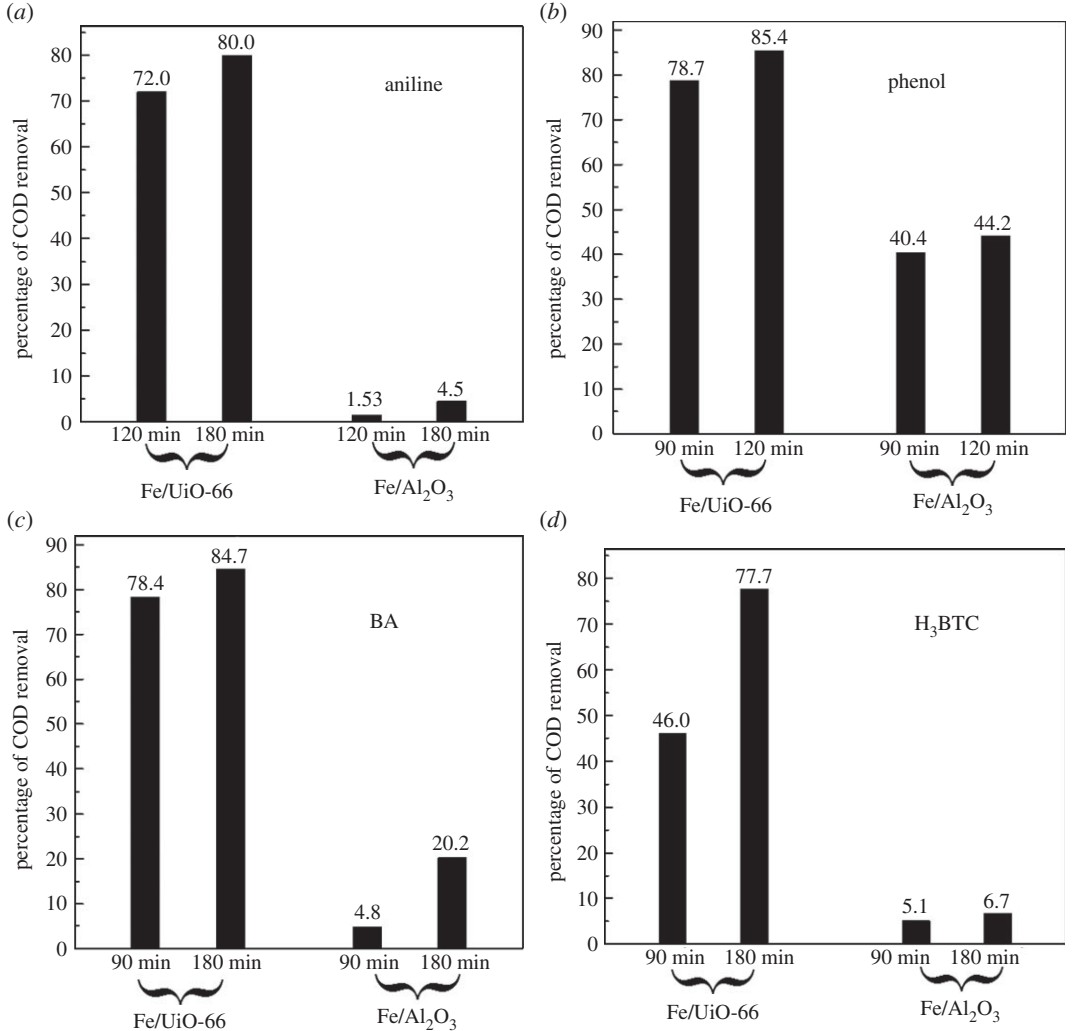

**Figure 8.** Catalytic removal of the COD value of various benzene derivatives polluted water at the presence of 0.0087 mol $l^{-1}$ $H_2O_2$ using 1000 mg $l^{-1}$ Fe/UiO-66 and Fe/$Al_2O_3$ as catalysts, respectively as indicated. Types of benzene derivatives were (a) 500 mg $l^{-1}$ aniline, (b) 500 mg $l^{-1}$ phenol, (c) 600 mg $l^{-1}$ benzoic acid (BA) and (d) 1000 mg $l^{-1}$ 1,2,3-benzenetricarboxylic acid (BTA), respectively. Reaction temperature was 50°C. Reaction time is indicated under the black columns in the figure.

$H_2O_2$ in water (pH = 3.5). About 70% of COD was removed during 30 min under the optimum operating conditions. Thus, inspection of the catalytic test parameters and the COD removal percentage further shows the superiority of the Fe/UiO-66 catalyst for the phenol removal.

Similar catalytic efficiency trends were also observed for the elimination of BA (figure 8c) and of $H_3BTC$ (figure 8d) from water in the cases of aniline and phenol. For example, for the elimination of BA, the COD removal percentage given by the Fe/UiO-66 catalyst was over 4 times (approx. 85%) that given by the Fe/$Al_2O_3$ catalyst at the reaction time of 180 min. Pariente et al. [45] have used 0.6 g $l^{-1}$ $Fe_2O_3$/SBA-15 to catalyse photo-Fenton oxidation (365 nm UV light + $H_2O_2$ with the 60% stoichiometric amount) of BA (50 mg $l^{-1}$). At the reaction time of 240 min, the percentage COD reduction was approximately 88%. With a comparable COD removal percentage, the Fe/UiO-66 catalyst in this work had the advantages of (i) much higher BA/catalyst ratio, and (ii) no need of light irradiation. These advantages reinforced the superiority of the Fe/UiO-66 catalyst for removing BA and $H_3BTC$ from water, and further improved its general applicability for removing benzene derivatives.

### 3.4.2. Acetone

Acetone is an important feedstock in the chemical industrial and also a typical organic pollutant. Most of the studies for acetone elimination focused on the oxidative combustion method, (e.g. [46]) while those for acetone removal from water are sparse [47]. Figure 9 shows the catalytic removal of acetone was more

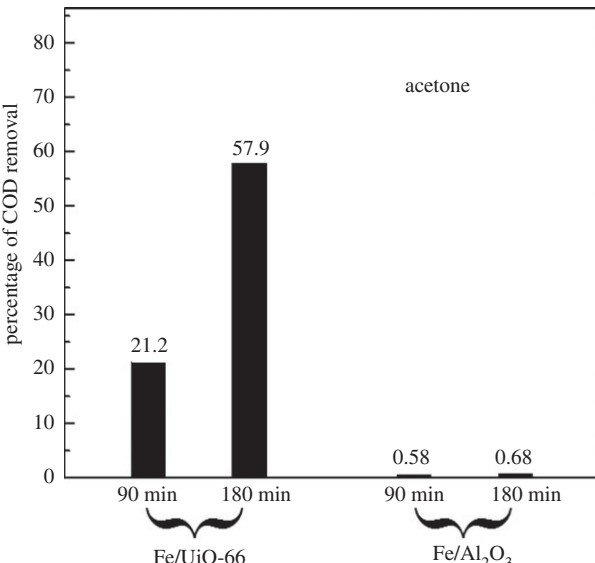

**Figure 9.** Catalytic removal of the COD value of 500 mg l$^{-1}$ aqueous acetone solution in the presence of 0.0087 mol l$^{-1}$ H$_2$O$_2$ using 1000 mg l$^{-1}$ Fe/UiO-66 and Fe/Al$_2$O$_3$ as catalysts, respectively, as indicated. Reaction temperature was 50°C. Reaction time is indicated under the black columns in the figure.

difficult than the removal of benzene derivatives using either Fe/UiO-66 or Fe/Al$_2$O$_3$ as the catalyst. However, a mild COD removal percentage can still be achieved by Fe/UiO-66 at the reaction time of 180 min. In contrast to the Fe/UiO-66 case, Fe/Al$_2$O$_3$ almost did not present notable catalytic for acetone removal from water.

To briefly summarize the proceeding results from the catalytic tests, the Fe/UiO-66 showed a general applicability and superior catalytic performances for catalytic removal of various organic compounds from water, including dyes, benzene derivatives and small-molecule organic compounds. In this context, the reason that the Fe/UiO-66 showed considerably higher catalytic activity than conventional Fe/Al$_2$O$_3$ for the tested reactions became an interesting issue. Although MOF in general and UiO-66 in particular has large surface area, Fe/UiO-66 did not present rather high BET surface area in this work (table 2). The BET surface area of Fe/UiO-66 was even lower than that of Fe/Al$_2$O$_3$, indicating that surface area was not a beneficial factor for the catalytic superiority of Fe/UiO-66. It is well known that Fe$^{3+}$ ion is the catalytic active species for homogeneous Fenton reactions [5–8]. Although the Fe components are also believed to be catalytically active for the oxidation of organic compounds by H$_2$O$_2$ through heterogeneous Fenton reactions [5–8], the active Fe structure in the molecular level really responsible for the catalytic process is still poorly understood. Fe/UiO-66 showing higher catalytic activity than Fe/Al$_2$O$_3$ implied that the former had a much larger amount of the 'active Fe structure'. In this sense, further systematic characterization of the chemical state of Fe on Fe/UiO-66 can help better understand the chemical nature of 'active Fe structure' for heterogeneous Fenton reactions.

## 4. Conclusion

This paper presents a novel preparation method for MOF-supported Fe catalyst, characterization results for Fe/UiO-66, and catalytic applications of Fe/UiO-66 for oxidative removal of organic compounds from water. The major findings are described as follows.

(1) A carrier-gas free VD method was established for preparation of MOF-supported Fe catalyst, and the Fe/UiO-66 catalyst was successfully prepared by using the VD method. To our knowledge, this paper provides the first case in terms of preparation of MOF-supported Fe catalysts using the vapour-based method.

(2) Under the present preparation conditions, a Fe loading of around 7.0–8.5 wt% can be obtained on Fe/UiO-66. The residence of the Fe components did not result in notable changes of the UiO-66 support in crystal structure (by XRD) and in morphology (by SEM), while it led to a significant

decrease in surface area and total pore volume. The thermal stability of Fe/UiO-66 had only a minor decrease compared to the excellent thermal stability of UiO-66.

(3) Fe/UiO-66 can efficiently catalyse the oxidative removal of MO by $H_2O_2$ in an aqueous solution, and UiO-66 simply adsorbed MO from the aqueous solution.

(4) Fe/UiO-66 was rather effective for catalytic removal of several benzene derivatives from water and moderately effective for removing acetone.

It is interesting that the superior catalytic performances of Fe/UiO-66 for organic pollutants removal were achieved by using near neutral solution instead of acidic solutions. The general applicability and superior catalytic performances in the oxidative removal of organic compounds made Fe/UiO-66 a hopeful catalyst for its environmental use.

Data accessibility. Data available from the Dryad Digital Repository at: https://doi.org/10.5061/dryad.k05cb25 [48].

Authors' contributions. H.Z. partly designed the study, prepared and characterized all of the supported Fe catalysts and prepared the draft of the manuscript. B.C. primarily developed the vapour-based method for preparing Fe catalysts, designed and established the method for catalytic reaction tests. W.C. helped develop the vapour-based method for preparing Fe catalysts, helped design and establish the method for catalytic reaction tests. Y.X. partly designed the study especially the part of catalyst preparation and catalyst characterization, helped in the preparation of the experimental set-ups, analysed part of the data and helped prepare and revise the manuscript. T.Y. helped designed the study, analysed part of the data and helped improve the manuscript. C.W. helped design the study, analysed part of the data and helped improve the manuscript. X.L.. primarily designed the whole study, helped analyse the data, and prepared and revised the manuscript.

Competing interests. We have no competing interests.

Funding. The financial support came from the National Natural Science Foundation of China (21576291, 21306230 and 21604094), Shandong Province Natural Science Foundation (ZR2014BM002, ZR2012BQ020 and ZR2017LB013) and the Fundamental Research Funds for the Central Universities (15CX05052A, 17CX02067 and 15CX08010A).

Acknowledgements. Y. Xi and X. Lin thank Dr L. Wei for providing part of the experimental set-ups used in this work, and some insightful advice. The authors are also grateful to two anonymous reviewers, who provided comments that improved the manuscript.

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
