## [Reviewer comments · Royal Society Open Science]

Review History

RSOS-182047.R0 (Original submission)

Review form: Reviewer 1

Is the manuscript scientifically sound in its present form?

Yes

Are the interpretations and conclusions justified by the results?

Yes

Is the language acceptable?

Yes

Is it clear how to access all supporting data?

Yes

Do you have any ethical concerns with this paper?

No

Have you any concerns about statistical analyses in this paper?

No

Recommendation?

Accept as is

Comments to the Author(s)

All the issues raised by the previous Reviewers have been properly replied and revised in the modified submission materials, thus I recommend it for publication in the Royal Society Open Science.

Review form: Reviewer 2

Is the manuscript scientifically sound in its present form?

No

Are the interpretations and conclusions justified by the results?

No

Is the language acceptable?

Yes

Is it clear how to access all supporting data?

Yes

Do you have any ethical concerns with this paper?

No

Have you any concerns about statistical analyses in this paper?

No

Recommendation?

Reject

Comments to the Author(s)

In this work, authors have reported synthesis of UO-66 supported Fe catalyst and also their application to oxidation of pollutants like methyl orange and aniline. Oxidation of organic pollutants with Fe-H₂O₂ combination is very well known. The Fe/UiO-66 material is poorly characterized. It is not clear, what is the state of Fe in the catalyst. Also some data interpretations are not logical. For example, BJH pore-size distribution analysis the authors conclude in table 2 that average pore diameter is 2.0. This does not make sense as the materials are clearly micro porous as evident from type-1 isotherm(Figure 5a). Moreover, UiO-66 can only have 0.75 nm and 1.2 nm pores from the single-crystal structure data reported in the literature. Further it is surprising to note just around 8 wt% Fe has brought down the porosity to 0.08 cm³/g. It was attributed to presence of Fe in the pores, which is also not logical. As simple maths will tell it is not possible. Also, close analysis of Fe/UiO-66 TGA data shows that the material decomposes starting from around 280 C, but materials were calcined at 300 C for BET measurements. This shows poorly conducted experiments. Also, most of the responses given to previous reviewers are also not scientifically sound. There are several other scientific drawbacks in this paper. Therefore I recommend rejection of this paper.

Review form: Reviewer 3

Is the manuscript scientifically sound in its present form?

Yes

Are the interpretations and conclusions justified by the results?

Yes

Is the language acceptable?

Yes

Is it clear how to access all supporting data?

Yes

Do you have any ethical concerns with this paper?

No

Have you any concerns about statistical analyses in this paper?

No

Recommendation?

Accept with minor revision (please list in comments)

Comments to the Author(s)

In this work, the authors reported a facile vapor deposition method to deposit catalytic Fe species on metal organic framework of UiO-66. The obtained materials showed good thermal stability above 200 degree C in air. More importantly, they showed that Fe/UiO-66 was effective to catalyze the oxidation of benzene derivatives like aniline in water in terms of chemical oxygen demand (COD) removal efficiency. The paper is well written and recommended for publication in Royal Society Open Science after addressing the following one important issue:

1. What is the reaction for the considerably higher activity of Fe/UiO-66 than conventional Fe/Al₂O₃ catalyst? More discussion might be needed.

Decision letter (RSOS-182047.R0)

25-Feb-2019

Dear Dr Lin:

Title: UiO-66 supported Fe catalyst: A vapor deposition preparation method and its superior catalytic performance for removal of organic pollutants in water

Manuscript ID: RSOS-182047

The editor assigned to your manuscript has now received comments from reviewers. We would like you to revise your paper in accordance with the referee and Subject Editor suggestions which

can be found below (not including confidential reports to the Editor). Please note this decision does not guarantee eventual acceptance.

Please submit your revised paper before 20-Mar-2019. Please note that the revision deadline will expire at 00.00am on this date. If we do not hear from you within this time then it will be assumed that the paper has been withdrawn. In exceptional circumstances, extensions may be possible if agreed with the Editorial Office in advance. We do not allow multiple rounds of revision so we urge you to make every effort to fully address all of the comments at this stage. If deemed necessary by the Editors, your manuscript will be sent back to one or more of the original reviewers for assessment. If the original reviewers are not available we may invite new reviewers.

RSC Associate Editor:
Comments to the Author:
(There are no comments.)

RSC Subject Editor:
Comments to the Author:
(There are no comments.)

Reviewers' Comments to Author:

Reviewer: 1

Comments to the Author(s)

All the issues raised by the previous Reviewers have been properly replied and revised in the modified submission materials, thus I recommend it for publication in the Royal Society Open Science.

Reviewer: 2

Comments to the Author(s)

In this work, authors have reported synthesis of UO-66 supported Fe catalyst and also their application to oxidation of pollutants like methyl orange and aniline. Oxidation of organic pollutants with Fe-H₂O₂ combination is very well known. The Fe/UiO-66 material is poorly characterized. It is not clear, what is the state of Fe in the catalyst. Also some data interpretations are not logical. For example, BJH pore-size distribution analysis the authors conclude in table 2 that average pore diameter is 2.0. This does not make sense as the materials are clearly micro porous as evident from type-1 isotherm(Figure 5a). Moreover, UiO-66 can only have 0.75 nm and 1.2 nm pores from the single-crystal structure data reported in the literature. Further it is surprising to note just around 8 wt% Fe has brought down the porosity to 0.08 cm³/g. It was attributed to presence of Fe in the pores, which is also not logical. As simple maths will tell it is not possible. Also, close analysis of Fe/UiO-66 TGA data shows that the material decomposes starting from around 280 C, but materials were calcined at 300 C for BET measurements. This shows poorly conducted experiments. Also, most of the responses given to previous reviewers are also not scientifically sound. There are several other scientific drawbacks in this paper. Therefore I recommend rejection of this paper.

Reviewer: 3

Comments to the Author(s)

In this work, the authors reported a facile vapor deposition method to deposit catalytic Fe species on metal organic framework of UiO-66. The obtained materials showed good thermal stability above 200 degree C in air. More importantly, they showed that Fe/UiO-66 was effective to catalyze the oxidation of benzene derivatives like aniline in water in terms of chemical oxygen demand (COD) removal efficiency. The paper is well written and recommended for publication in Royal Society Open Science after addressing the following one important issue:

1. What is the reaction for the considerably higher activity of Fe/UiO-66 than conventional Fe/Al₂O₃ catalyst? More discussion might be needed.

Author's Response to Decision Letter for (RSOS-182047.R0)

See Appendix A.

Decision letter (RSOS-182047.R1)

19-Mar-2019

Dear Dr Lin:

Title: UiO-66 supported Fe catalyst: A vapor deposition preparation method and its superior catalytic performance for removal of organic pollutants in water

Manuscript ID: RSOS-182047.R1

It is a pleasure to accept your manuscript in its current form for publication in Royal Society Open Science. The chemistry content of Royal Society Open Science is published in collaboration with the Royal Society of Chemistry.

RSC Associate Editor
Comments to the Author:
(There are no comments.)

Reviewer(s)' Comments to Author:

Appendix A

Response to Referees

Reviewers' Comments to Author:

Reviewer: 1

Comments to the Author(s)

All the issues raised by the previous Reviewers have been properly replied and revised in the modified submission materials, thus I recommend it for publication in the Royal Society Open Science.

Response: Thanks for your time and effort in the whole review process, and thanks a lot for the recommendation.

Reviewer: 2

Comments to the Author(s)

In this work, authors have reported synthesis of UiO-66 supported Fe catalyst and also their application to oxidation of pollutants like methyl orange and aniline. Oxidation of organic pollutants with Fe-H₂O₂ combination is very well known.

The Fe/UiO-66 material is poorly characterized. It is not clear, what is the state of Fe in the catalyst.

Response: Thanks for comment. Actually and obviously we have employed most of the common used catalyst characterization techniques for our Fe/UiO-66 catalyst, such as XRD, N₂-adsorption, SEM, EDAX mapping, TG, etc. All the characterization results help people to better understand the catalyst structure and the related preparation process. It should be noted that since this paper emphasis the novelty of preparation for MOF supported Fe catalyst, only essential and necessary characterization techniques were employed presently. Generally speaking there are always a lot of techniques that can provide helpful information for a catalyst. The scientific story never ends, but a paper has an end. The key point is

what the main purpose of this paper is. For the instance of the issue of "state of Fe", in principle, it may encompass the following aspects such as, (i) oxidation state (ii) aggregation state, (iii) coordination state, (iv) distribution of surrounding atoms (v) exposure state to reactant, etc. Each of the above information is important to understand a catalytic process. We believe the story of state of Fe itself is definitely enough to write a long paper even several paper. We admit that it is an interesting story, however, it is obviously beyond the scope of this paper.

Further explanation can be seen in a paragraph added in the revised manuscript. See words from the third line to the end of this paragraph in Page 22 in the revised manuscript.

End of reply.

Also some data interpretations are not logical. For example, BJH pore-size distribution analysis the authors conclude in table 2 that average pore diameter is 2.0. This does not make sense as the materials are clearly micro porous as evident from type-1 isotherm(Figure 5a). Moreover, UiO-66 can only have 0.75 nm and 1.2 nm pores from the single-crystal structure data reported in the literature.

Response: Thanks for the comment. Actually there was a sharp peak around 0.75 nm (Fig. 5c) and a rather small hump around 1.2 nm. The purchased UiO-66 was powder so as it may not perfectly single-crystal, and there can have some contribution of mesopores like inner cracked spaces. The XRD results showed the original UiO-66 was OK. A material having average pore diameter of 2.0 nm may not necessarily mesoporous. It is reasonable to present a type-I isotherm.

End of reply.

Further it is surprising to note just around 8 wt% Fe has brought down the porosity to 0.08 cm³/g. It was attributed to presence of Fe in the pores, which is also not logical. As simple maths will tell it is not possible.

Response: Thanks for comment. We agree simple maths really helps clarify this issue. However, the conclusion is actually reverse to the one given by the reviewer. The deduction of the above comment is not logical! Recall the exclusive method.

The pure UiO-66 had large surface area and pore volume before and after calcination. Suppose the Fe component resided on the outer surface of UiO-66, or just physically mixed with UiO-66, the 8 wt% contribution of Fe was not possible to have a great impact on the overall BET surface or porosity.

So actually it is the only logical conclusion to deduce that the Fe component was in the pores instead of in other places through simple maths. It should be noticed that the BET surface area and the porosity were not the ideal geometrical value, but calculated from the number of adsorbates (N₂). Considering the pores in UiO-66 (e.g. ~0.75 nm) were rather small, the occupation of the micropores by the Fe components resulted in a sharply decreased space for N₂, leading to a sharp decreased in the measured porosity. The EDX mapping (Fig. 4) results showed that the Fe distribution of Fe/UiO-66 was well consistent with the Zr distribution at a same region. Since the Zr distribution was determined by the UiO-66 framework, the EDX mapping results further support the above hypothesis.

End of reply.

Also, close analysis of Fe/UiO-66 TGA data shows that the material decomposes starting from around 280 C, but materials were calcined at 300 C for BET measurements. This shows poorly conducted experiments.

Response: Thanks for comment. Actually the pure UiO-66 instead of Fe/UiO-66 was calcined at 300 °C before the vapor deposition. Before further characterization, the catalyst was calcined at 200 °C instead of 300 °C (see the Catalyst Preparation section in Page 6 in the manuscript). So there is no problem for our experiment.
End of reply.

Also, most of the responses given to previous reviewers are also not scientifically sound. There are several other scientific drawbacks in this paper.

Therefore I recommend rejection of this paper.

Reviewer: 3

Comments to the Author(s)

In this work, the authors reported a facile vapor deposition method to deposit catalytic Fe species on metal organic framework of UiO-66. The obtained materials showed good thermal stability above 200 degree C in air. More importantly, they showed that Fe/UiO-66 was effective to catalyze the oxidation of benzene derivatives like aniline in water in terms of chemical oxygen demand (COD) removal efficiency. The paper is well written and recommended for publication in Royal Society Open Science after addressing the following one important issue:

1. What is the reaction for the considerably higher activity of Fe/UiO-66 than conventional Fe/Al₂O₃ catalyst? More discussion might be needed.

Response: Thanks for your time and effort in the whole review process. Thanks for the suggestion. We guess the above-mentioned word of "reaction" can be "reason". We have added a paragraph for further discussion on the reason for higher activity of Fe/UiO-66 than conventional Fe/Al₂O₃ catalyst.

We added the following words of discussion.

In the mean time the reason for the considerably higher catalytic activity of Fe/UiO-66 than conventional Fe/Al₂O₃ for the tested reactions became an interesting issue. Although MOF in general and UiO-66 in particular has large surface area, Fe/UiO-66 did not present rather high BET surface area in this work (see Table 2). The BET surface area of Fe/UiO-66 was even lower than that of Fe/Al₂O₃, indicating that surface area was not a beneficial factor for the catalytic superiority of Fe/UiO-66. It is well known that Fe³⁺ ion is the catalytic active species for homogenous Fenton reactions. Although the Fe components are also believed to be catalytic active for the oxidation of organic compounds by H₂O₂ through heterogeneous Fenton reactions, the active Fe structure in the molecular level really responsible for the catalytic process is still poor understood. Fe/UiO-66 showing higher catalytic activity than Fe/Al₂O₃ implicated that the former had much larger amount of the “active Fe structure”. In this sense, further systematic characterization of the chemical state of Fe on Fe/UiO-66 can help better understanding the chemical nature of “active Fe structure” for heterogeneous Fenton reactions.

These words can be seen in the paragraph before “Conclusion” section in Page 22 in the revised manuscript.

End of reply.